# The interplay of seizures-induced axonal sprouting and transcription-dependent *Bdnf* repositioning in the model of temporal lobe epilepsy

Anna Skupien-Jaroszek[1☯], Agnieszka Walczak[1☯], Iwona Czaban[1], Katarzyna Karolina Pels[1], Andrzej Antoni Szczepankiewicz[1], Katarzyna Krawczyk[1], Błażej Ruszczycki[1], Grzegorz Marek Wilczynski[1†], Joanna Dzwonek[1]*, Adriana Magalska[1,2]*

**1** Laboratory of Molecular and Systemic Neuromorphology, Nencki Institute of Experimental Biology, Polish Academy of Sciences, Warsaw, Poland, **2** Laboratory of Molecular Basis of Cell Motility, Nencki Institute of Experimental Biology, Polish Academy of Sciences, Warsaw, Poland

☯ These authors contributed equally to this work.
† Deceased.
* a.magalska@nencki.edu.pl (AM); j.dzwonek@nencki.edu.pl (JD)

**Data Availability Statement:** All relevant data are within the manuscript and its Supporting Information files.

## Abstract

The Brain-Derived Neurotrophic Factor is one of the most important trophic proteins in the brain. The role of this growth factor in neuronal plasticity, in health and disease, has been extensively studied. However, mechanisms of epigenetic regulation of *Bdnf* gene expression in epilepsy are still elusive. In our previous work, using a rat model of neuronal activation upon kainate-induced seizures, we observed a repositioning of *Bdnf* alleles from the nuclear periphery towards the nuclear center. This change of *Bdnf* intranuclear position was associated with transcriptional gene activity. In the present study, using the same neuronal activation model, we analyzed the relation between the percentage of the *Bdnf* allele at the nuclear periphery and clinical and morphological traits of epilepsy. We observed that the decrease of the percentage of the *Bdnf* allele at the nuclear periphery correlates with stronger mossy fiber sprouting—an aberrant form of excitatory circuits formation. Moreover, using *in vitro* hippocampal cultures we showed that *Bdnf* repositioning is a consequence of transcriptional activity. Inhibition of RNA polymerase II activity in primary cultured neurons with Actinomycin D completely blocked *Bdnf* gene transcription and repositioning occurring after neuronal excitation. Interestingly, we observed that histone deacetylases inhibition with Trichostatin A induced a slight increase of *Bdnf* gene transcription and its repositioning even in the absence of neuronal excitation. Presented results provide novel insight into the role of BDNF in epileptogenesis. Moreover, they strengthen the statement that this particular gene is a good candidate to search for a new generation of antiepileptic therapies.

**Funding:** JD was supported by the Polish National Science Centre grant No 2015/17/B/NZ4/02540, AS was supported by the Polish National Science Centre grant No 2018/29/B/NZ4/01473, AM was supported by the Polish National Science Centre grant No UMO-2015/18/E/ NZ3/00730. AAS was supported by the Polish National Science Centre grant No 2014/15/N/NZ3/04468. KKP was partially supported by the ETIUDA grant from the Polish National Science Centre no. UMO-2019/32/T/NZ4/00502. https://www.ncn.gov.pl The funders had no role in study design, data collection and analysis, decision to publish, or preparation of the manuscript.

**Competing interests:** The authors have declared that no competing interests exist.

## Introduction

The Brain-Derived Neurotrophic Factor (BDNF) is one of the most important neurotrophins in the brain. Acting *via* its synaptic receptor Tropomyosin-related kinase B (TrkB), BDNF is involved in neuronal differentiation, survival, and synaptic plasticity [1–3]. Thus, it plays an important role in the number of neurological and psychiatric disorders such as Parkinson's disease [4], schizophrenia [5], depression [6], bipolar disease [7], and epilepsy [8–10].

Currently, it is known that BDNF is involved in the event of aberrant synaptic plasticity called mossy fiber sprouting, observed in temporal lobe epilepsy (TLE), which is one of the most common types of epilepsy in adults [11]. The level of both BDNF protein [12] and mRNA [13–15] were described to be elevated after seizures in the temporal lobe and hippocampi of epileptic patients. Experiments performed on animals and *in vitro* models showed that BDNF causes hypertrophy of granule neurons [16] and increased mossy fiber branching [17]. Moreover, intrahippocampal infusion of BDNF induced mild seizures with the development of mossy fiber sprouting [18]. Those findings support the involvement of BDNF in the aberrant synapse formation in TLE, however, underlying molecular mechanisms are still not clear.

The current trend in neuroscience is to look for mechanisms of neuronal functioning at a gene expression level. BDNF encoding gene is a so-called delayed, immediate-early gene induced in the later phase of neuronal activation [19]. It consists of 9 exons differentially expressed in humans, mice, and rats [20]. Regulation of BDNF expression after neuronal excitation has been quite well understood at the level of transcription factors and chromatin modifications. It is known that the 3' end of the protein-coding exon is spliced to one of the eight of 5' exons, which are controlled by their unique promoters [20–22]. Moreover, *Bdnf* can be epigenetically down-regulated through DNA methylation and histone deacetylation. The aforementioned epigenetic changes result in the recruitment of REST/NRSF complex [23] and MeCP2 [24,25] and chromatin remodeling. Conversely, the up-regulation of the gene can be triggered by DNA demethylation and/ or histone acetylation, which was already presented in both *in vitro* [20] and *in vivo* studies [26]. Importantly, the level of histone H3 acetylation at the *Bdnf* promoters IV and VI may underlie sustained up-regulation of transcription following chronic electroconvulsive shock [27]. Fukuchi et al. [28] showed that valproic acid, an inhibitor of histone deacetylases commonly used antiepileptic drug, increases expression of *Bdnf* under control of promoter I.

Studies of the last decade have shown that the genome is spatially organized within the nucleus [29]. Rearrangements of chromatin are involved in the regulation of gene expression, and the radial position of genes reflects their expression [30]. In differentiated cells, the nuclear periphery is a repressive environment, where heterochromatin is recruited to the nuclear lamina [31]. Artificial localization of gene at the nuclear periphery, by tethering to the inner nuclear membrane, is sufficient to induce silencing of its expression [32]. The role of chromatin structure in the regulation of gene expression in neurons remains still unexplored. Crepaldi and colleagues [33] showed that activity-dependent genes, including *Bdnf*, were repositioned to transcription factories after KCl induced depolarization in cultured cortical neurons. Moreover, in our previous studies [34] we showed that during neuronal excitation and epileptogenesis *Bdnf* alleles had been detached from the nuclear lamina and repositioned from the nuclear periphery toward the nuclear center. The observed phenomenon was associated with changes in *Bdnf* expression. However, it was not clear whether *Bdnf* repositioning had been a cause or a consequence of the gene transcriptional activity. Therefore, in the current study, we are addressing this interesting question.

## Materials and methods

### Animals

The experiments were performed on young, adult male Wistar rats, weighing 170–250 g, obtained from Mossakowski Medical Research Centre, Polish Academy of Sciences. Animals were kept under a 12 h light/dark cycle, with unlimited food and water supplies. All procedures were performed with the consent of the 1st Local Ethical Committee in Warsaw (Permission number LKE 306/2017).

### Induction of seizures

Seizures were evoked by two doses of kainate (5 mg/kg, Sigma-Aldrich) (0.5% solution in saline, pH 7), administered intraperitoneally in 1 h intervals, and scored as described by Hellier et al [35]. The animals were taken for further studies regardless of whether they fulfilled the criterion of the full status epilepticus, or not [35]. To reduce kainate-induced mortality (up to 10%), diazepam (25mg/kg) was administrated intraperitoneally 3–6 hours after seizure onset. To monitor animal health, the rats were constantly observed by experimenters, for 6 hours after kainate injection, and 4 hours/day in the subsequent 4 weeks. All animals were euthanized with an overdose of pentobarbital sodium (Nembutal 150 mg/kg, i.p.) and perfused with 4% paraformaldehyde in PBS at the end of the experiments, and the brain tissues were collected.

### Estimation of clinical and morphological traits

Clinical and morphological traits such as the intensity of seizures, ruffling of the fur, forelimb clonus, body, and head tremor, losing posture, immobility and aggression have been estimated for 6 hours after kainate injection, and 4 hours/day in subsequent 4 weeks. The intensity of seizures upon kainate treatment was scored according to the 6-grade modified Racine's scale (0-lack of seizures to 5- fully developed tonic-clonic seizures with a loss of posture) [35]. Ruffling of the fur was scored on a 3-grade scale (0-lack of ruffling to 2-extensively ruffled fur). Forelimb clonus was scored according to a 7-grade scale (0-lack of clonus to 6-very strong forelimb clonus). Body and head tremor was scored according to a 6-grade scale (0-lack of tremor to 5-very strong tremor). Loss of posture was scored according to the 4-grade scale (0-lack of loss of posture to 4 constant loss of posture). Immobility was scored according to the 10-grade scale (0-completely active to 9-lack of activity). Aggression was scored on a 4-grade scale (0-lack of aggression to 3-extreme aggression). To minimize the bias, two independent observers performed the scoring of clinical parameters. Mossy fiber sprouting was verified by immunofluorescent staining for synaptoporin and scored on a 5-grade scale (0-lack of sprouting to 4-very strong sprouting). The mean synaptoporin intensity was measured for each individual using ImageJ. Measurements were done on confocal images in a specific area of the molecular layer of the dentate gyrus depicted in Fig 2B. Additionally, two independent observers estimated sprouting visually in the same brain area directly using a microscope.

### Primary neuronal hippocampal cultures and treatment

Primary neuronal cultures were prepared from the hippocampi of P0 rat brains as described previously [36]. Chemical long-term potentiation (cLTP) was initiated by stimulating the cells for 2 hours with 50 μM of picrotoxin, 50 μM forskolin, and 0.1 μM rolipram (all from Sigma-Aldrich). Transcription was inhibited by 2h exposure to 8mg/ml of Actinomycin D (Sigma-Aldrich). Histone deacetylases were inhibited by 12h treatment with 200 nM Trichotstatin A (Abcam) or 250 nM romidepsin (Cayman Chemical Company) for 2h.

## Fluorescent *in situ* hybridization

Fluorescent *in situ* hybridization was performed according to the protocol of Cremer et al. (2008) [37] on 30 μm-thick brain cryosections of the 4% paraformaldehyde-perfused, 8 kainate-treated, and 4 control animals, as well as on 3 independent primary hippocampal neuronal cultures fixed with 4% paraformaldehyde at DIV 14 (day *in vitro*). As templates for *Bdnf* FISH probes, CH230-449H21 BAC obtained from Children's Hospital Oakland Research Institute were used. Probes were verified on rat metaphase spreads. The probes were labeled using the standard nick-translation procedure. Biotinylated probes were detected using Alexa Fluor 488- conjugated avidin (Invitrogen), followed by FITC-conjugated rabbit anti-avidin antibody (Sigma-Aldrich).

## Immunostaining

Immunostaining for synaptoporin was performed on 30 μm-thick brain cryosections of the 4% paraformaldehyde-perfused animal using standard immunofluorescent staining protocol [38]. 1 μg/ml of rabbit polyclonal anti synaptoporin (Synaptic Systems) antibody was used. The intensity of immunostaining was estimated using the 5-grade scale (0-no staining to 4-strong staining). The neuronal damage was examined by staining with Fluoro-Jade B (Millipore) according to the method of Schmued et al [39].

## Image acquisition

Fluorescent specimens were examined under TCS SP8 confocal microscope (Leica) or Zeiss 800 confocal microscope (Zeiss), by sequential scanning of images, with a pixel size of 80 nm and axial spacing of 210 nm, using a PlanApo oil-immersion 63 (1.4 numerical aperture) objective.

## Quantitative image analysis

The minimal distance from the nuclear periphery and *Bdnf* alleles in neuronal nuclei of animals was calculated using custom-written software, Segmentation Magick, described in Ruszczycki et al. [40]. For neuronal cultures, custom-written software Partseg was used, described in [41]. At least 120 nuclei, from 3 independent experiments were analyzed for each experimental variant.

## Real-time reverse transcriptase-PCR for Bdnf mRNA

Total cellular RNA was isolated from three independent hippocampal primary cultures using RNeasy Mini Kit (Qiagen) according to the manufacturer's procedure. 1 μg of RNA was subjected to RT reaction using the SuperScript first-strand synthesis system for RT-PCR (Invitrogen) according to the manufacturer's protocol. PCR was performed using SYBR Green PCR Master Mix (Thermo Fischer Scientific). Forward and reverse primers, were respectively: 5-CCATAAGGACGCGGACTTGTAC and 5-AGACATGTTTGCGGCATCCAGG.

## Statistical analysis

The correlation between sprouting intensity and percentage of BDNF alleles on the nuclear border was calculated with Anaconda Software Distribution (2020) (Anaconda Inc) retrieved from https://docs.anaconda.com/, using SciPy library.

The data for RT-PCR experiments were obtained from 3–5 independent batches of neurons and normalized to the control in each experiment. Statistical analysis of normalized data was performed using the Kruskal-Wallis group comparison, and the One-Sample T-test or Welch

Two Sample t-test for pairwise comparison, with the use of R [42] retrieved from https://docs.anaconda.com/.

For the FISH experiments, the differences in the percentage of the gene at the nuclear periphery were analyzed using chi-square (all 4 groups together) and Fisher's exact test for post hoc pairwise comparison. Since the contingency tables show precise counts, error bars were calculated as a standard deviation from the binomial distribution. The details are given in the Supplementary S1 File. The analysis was performed using GraphPad Prism software.

## Results

### Morphological traits of the kainate model of TLE and clinical implications of *Bdnf* repositioning in the hippocampal granule neurons

To verify the kainate model of TLE in rats, animals were carefully observed for 6 hours after a kainate injection, and 4 hours/day in the subsequent 4 weeks. The intensity of seizures has been assessed according to modified 6-grade Racine's scale (0-lack of seizures- 5- fully developed tonic-clonic seizures with a loss of posture). Additionally, clinical traits such as fur ruffling, forelimb tonus, body and head tremor, loss of posture, immobility, and aggression have been rated. Moreover, hippocampal specimens from the aforementioned animals were analyzed for neuronal damage in the DG granule cell layer and CA3 pyramidal layer by Fluoro-jade B staining and mossy fiber sprouting in DG molecular layer by synaptoporin staining.

In our model, 60% of animals underwent *status epilepticus* (fully developed tonic-clonic seizures with a loss of posture, Fig 1A and Table 1 and S1 File), and the remaining animals showed moderate seizure symptoms like head tremors (wet dog shaking) and fur ruffling. In 4 weeks following administration of kainate 5 out of 10 animals showed fur ruffling, 5 out of 10 demonstrated forelimb tonus, all animals exhibited body and head tremor, 5 out of 10 showed loss of posture, 7 out of 10 –immobility, and 5 out of 10- aggression (for details see Table 1).

Fluoro-jade B staining showed no neuronal damage in the DG granule cell layer and extensive cell death in the CA3 pyramidal layer in all animals (Fig 1B). Extend of mossy fiber

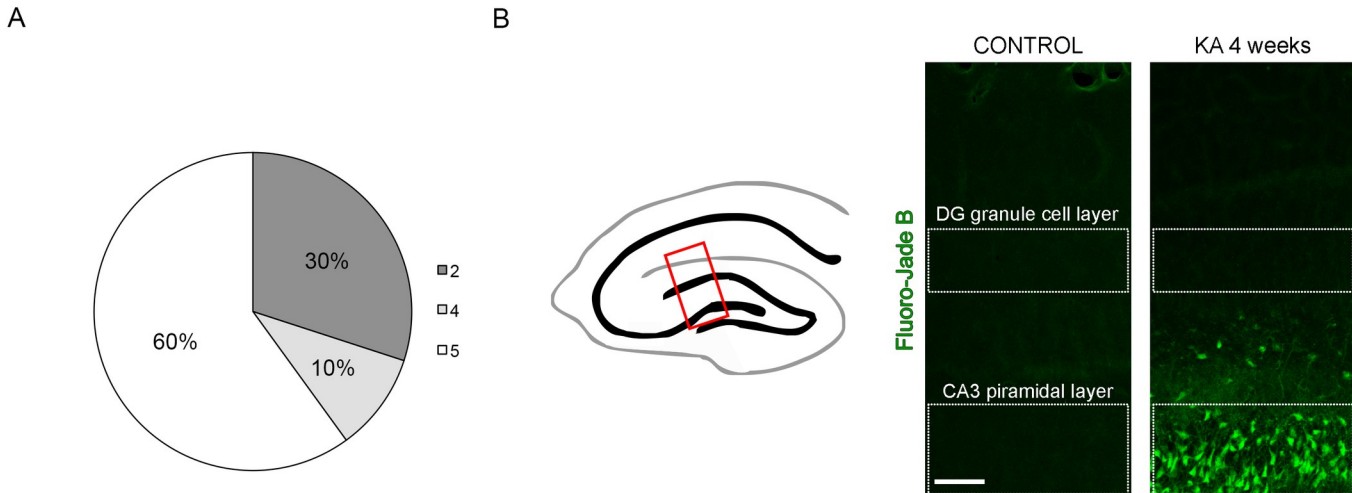

**Fig 1. Kainate induced seizures and neuronal damage.** A) Percentage of animals showing seizures after kainite administration. The intensity of seizure was scored according to the 6-grade modified Racine's scale (0-lack of seizures—5- fully developed tonic-clonic seizures with loss of posture). B) Fluoro-jade B staining (depicted in green) showing neuronal damage in the CA3 region, but not the DG region of the hippocampus of kainite treated animals. Scale Bar: 100 μm.

**Table 1. Estimation of the clinical and morphological traits in the kainate model of TLE.**

| animal ID | 1B | 1C | 1G | 2B | 2C | 2G | 3G | 3R | 4G | 4R |
|---|---|---|---|---|---|---|---|---|---|---|
| seizures (0–5) | 2 | 5 | 5 | 5 | 4 | 5 | 2 | 5 | 5 | 2 |
| % of nuclei with *Bdnf* at the perifery | 43 | 53 | 29 | 29 | 53 | 55 | 65 | 33 | 24 | 50 |
| sprouting level (0–4) | 2 | 0 | 4 | 2 | 1 | 3 | 0 | 1 | 4 | 1 |
| ruffling of fur (0–2) | 0 | 1 | 1 | 0 | 0 | 0 | 0 | 2 | 1 | 1 |
| forelimb tonus (0–6) | 0 | 0 | 0 | 1 | 0 | 3 | 1 | 6 | 2 | 0 |
| body/head tremor (0–5) | 3 | 4 | 1 | 2 | 1 | 3 | 2 | 4 | 5 | 2 |
| loss of posture (0–4) | 2 | 3 | 0 | 2 | 1 | 0 | 0 | 0 | 0 | 1 |
| inactivity (0–9) | 1 | 0 | 1 | 1 | 0 | 0 | 1 | 7 | 1 | 1 |
| aggression (0–3) | 1 | 0 | 1 | 0 | 0 | 1 | 0 | 0 | 3 | 1 |

For all traits, scales are shown in brackets (see Materials and Methods for full details).

sprouting was estimated by the intensity of synaptoporin immunostaining (Figs 2A and 2B and S1). In 8 out of 10 animals, it was possible to distinguish mossy fibers stained against

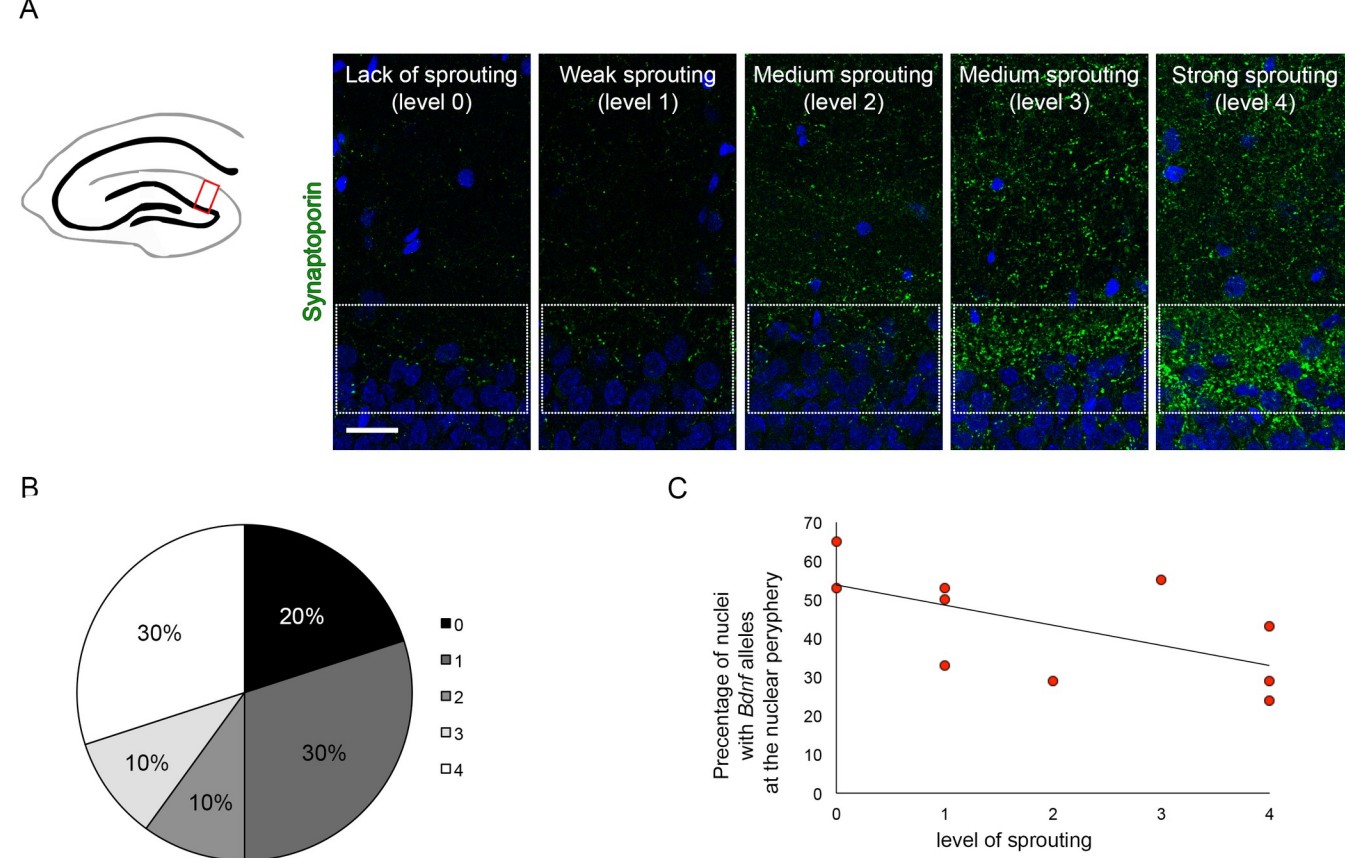

**Fig 2. Correlation between the intensity of sprouting and the percentage of the nuclei with *Bdnf* alleles at the nuclear periphery.** A) Mossy fiber sprouting was verified by immunofluorescent staining for synaptoporin in the molecular layer of DG region of the hippocampus. Representative pictures of different levels of sprouting of animals at 4 weeks after administration of kainate are shown. B) Percentage of animals showing different levels of synaptoporin staining intensity scored in 5-grade scale (0-lack of sprouting- 4 very strong sprouting). C) Correlation between the levels of sprouting in the DG, 4 weeks from the administration of kainate, measured in 5- grade scale, and percentage of the nuclei with *Bdnf* alleles localized at the nuclear periphery. Scale Bar: 100 μm.

synaptoporin (scored from 0- lack of sprouting, to 1—weak sprouting to 4—very strong sprouting, in Figs 2A and 2B and S1 and Table 1), where 30% showed moderate to very strong sprouting (scored ≥3). The level of sprouting was independent of the intensity of the initial seizures (right after kainate injection). All tested animals showed behavioral and morphological traits of epilepsy within 4 weeks from kainite treatment.

In our previous work [34] using the same model of neuronal activation upon kainate-induced seizures, we have observed repositioning of the *Bdnf* alleles in hippocampal granule neurons. We have shown that transcriptionally inactive *Bdnf* is attached to a nuclear lamina and localized at the nuclear periphery, after neuronal activation it has repositioned towards the nuclear center. Here we observed, that the level of sprouting correlated (R = -0.67, Pearson correlation) with the percentage of the nuclei with *Bdnf* allele localized at the nuclear periphery (Fig 2C). In animals with the strongest level of sprouting, in less than 50% of nuclei *Bdnf* alleles were present at the nuclear rim, while lack of sprouting correlated with the higher percentage of nuclei showing *Bdnf* localized in proximity to the nuclear envelope. We have found the same correlation calculated for sprouting levels estimated visually by two independent observers (R = -0.70) Moreover, the intensity of sprouting correlated positively with the level of aggression (R = 0,75, Pearson correlation), animals with a higher level of sprouting tend to be more aggressive (supplementary S1 File). Unfortunately, we haven't found any other correlations of behavioral traits neither with the level of sprouting nor the percentage of nuclei with *Bdnf* alleles on the nuclear periphery.

## The causal relationship between *Bdnf* allele transcriptional activity and repositioning

To further investigate the cause of the intranuclear reposition of *Bdnf* and its relationship with BDNF transcription, we used an *in vitro* model of neuronal excitation based on hippocampal dissociated cultures and a chemical model of long-term potentiation (cLTP) [43–45]. cLTP evoked by picrotoxin, forskolin, and rolipram was proven to be non-toxic for neurons and to induce a program of gene expression similar to the one observed in a brain upon stimulation. We observed that 2 hours after the initiation of long-term potentiation, *Bdnf* expression was significantly (4 times) increased compared to the control (Fig 3A). At the same time, the *Bdnf* allele had repositioned toward the center of a cell nucleus (Figs 3B–3D and S2). In the control cells (treated with a solvent alone) the *Bdnf* alleles were most frequently positioned at a nuclear margin, with 76.4% of alleles located 350 nm or less from a nuclear border (Fig 3C and 3D, blue bars). This distance is an approximate microscope resolution limit, hence it has been chosen as an indicator of allele proximity to the nuclear border, as previously described [34]. In activated neurons, we observed a distinct repositioning of the *Bdnf* gene from the nucleus periphery towards the nucleus center (Fig 3C and 3D, orange bars). The percentage of *Bdnf* alleles localized closer than 350 nm to the nuclear margin was significantly lower than in the control group (62.9% Fisher's exact tests). To verify whether *Bdnf* allele repositioning is a cause or a consequence of transcriptional activity, we performed experiments with the use of Actinomycin D, a potent inhibitor of RNA polymerase II. Preincubation with Actinomycin D for 2 hours was sufficient to inhibit *Bdnf* expression upon stimulation with cLTP (6 fold decrease compared to cLTP treatment, Fig 3A). Inhibition of transcription completely blocked *Bdnf* repositioning upon cLTP treatment (Figs 3B–3D and S2. The percentage of *Bdnf* alleles located at the nuclear periphery upon Actinomycin D and cLTP treatment was similar to control cells. Presented results show that transcriptional activity is a cause of *Bdnf* repositioning.

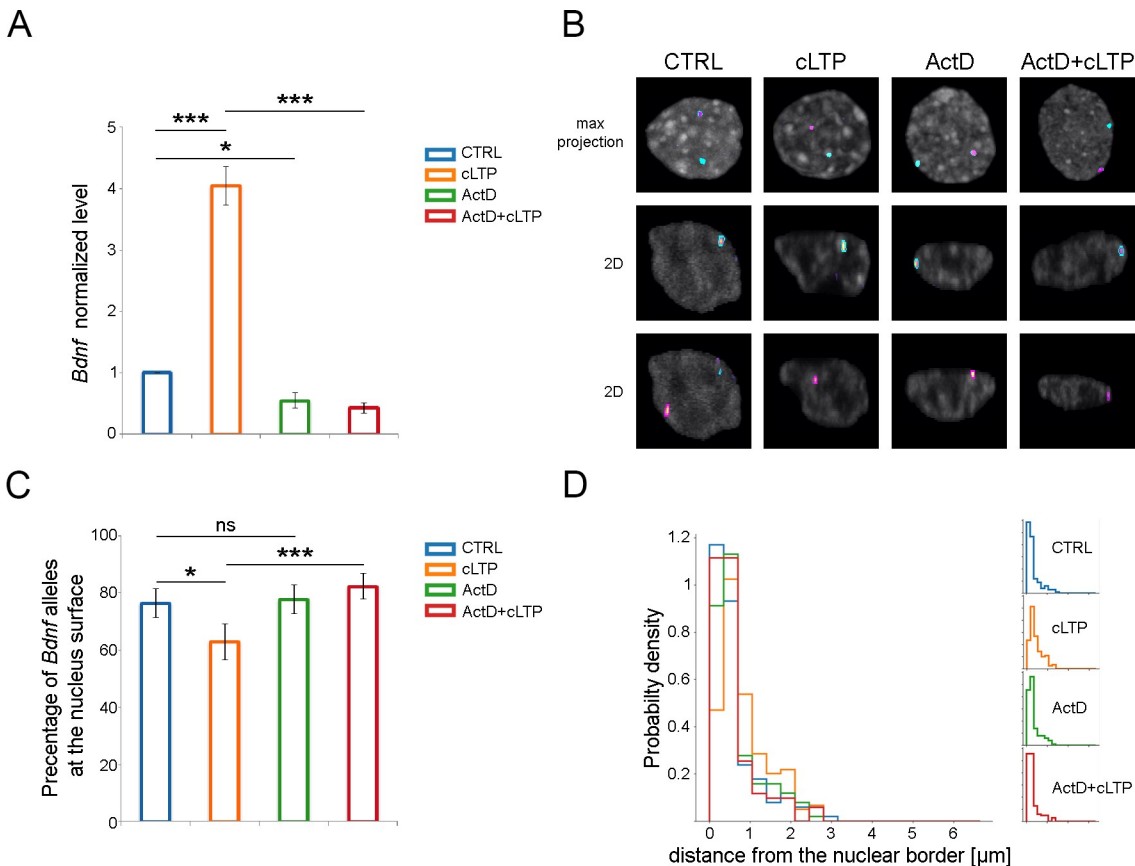

**Fig 3. The causal relationship between *Bdnf* transcriptional activity and *Bdnf's* allele repositioning.** (A) The graph shows the expression of *Bdnf* normalized to control in the hippocampal neurons incubated for 2 hours with DMSO vehicle (CTRL, blue bar) or picrotoxin, forskolin, and rolipram (cLTP, orange bar), incubated for 2 hours with Actinomycin D and 2 hours with DMSO (ActD, green bar) or picrotoxin, forskolin, and rolipram (ActD+cLTP, red bar). Kruskal-Wallis group comparison: p<0.01, and the One-Sample T-test or Welch Two Sample t-test for pairwise comparison: * p<0.05, ** p<0.01, *** p<0.001; error bars indicate standard error of the mean for 3 independent experiments (B) Representative picture of the nuclei of hippocampal neurons treated as described above. Hoechst 3342 staining for chromatin is shown in greyscale and segmentation of FISH signals for the *Bdnf* gene are shown in magenta and cyan. (C) Percentages of *Bdnf* alleles localized < 350 nm to the nuclear surface are shown (Chi-square test, all groups, p<0.001; Fisher's exact tests * p<0.05, *** p<0.001, error bars indicate standard deviation of the binomial distribution). (D) Quantitative analysis of the intracellular positions of *Bdnf* alleles in the nuclei of hippocampal neurons treated and color-coded as in A. The minimal distance between the respective alleles and nucleus surface is presented in the normalized histogram.

## Histone deacetylases are necessary for the attachment of *Bdnf* alleles to the nuclear lamina

Covalent modifications of chromatin were shown to be involved in the regulation of *Bdnf* gene expression [27,46,47]. Data indicate, that histone deacetylases (HDAC) play an important role in this process [48,49]. Therefore, we have raised a question of HDAC involvement in *Bdnf* repositioning observed after neuronal stimulation (Fig 4). Preincubation of hippocampal cultures with Trichostatin A, which is a commonly used HDAC paninhibitor, slightly increased *Bdnf* transcription (Fig 4A) and was sufficient to induce *Bdnf* allele repositioning toward the nuclear center (Figs 4B–4D and S3). The percentage of *Bdnf* alleles localized at the nuclear periphery was significantly lower upon TSA treatment compared to the control group. Induction of cLTP after Trichostatin A treatment did not decrease the percentage of alleles located at the nuclear periphery, in comparison to cLTP or TSA alone. The preincubation with a more specific HDAC inhibitor- romidepsin (Fig 5), selective for HDAC1 and HDAC2 deacetylases

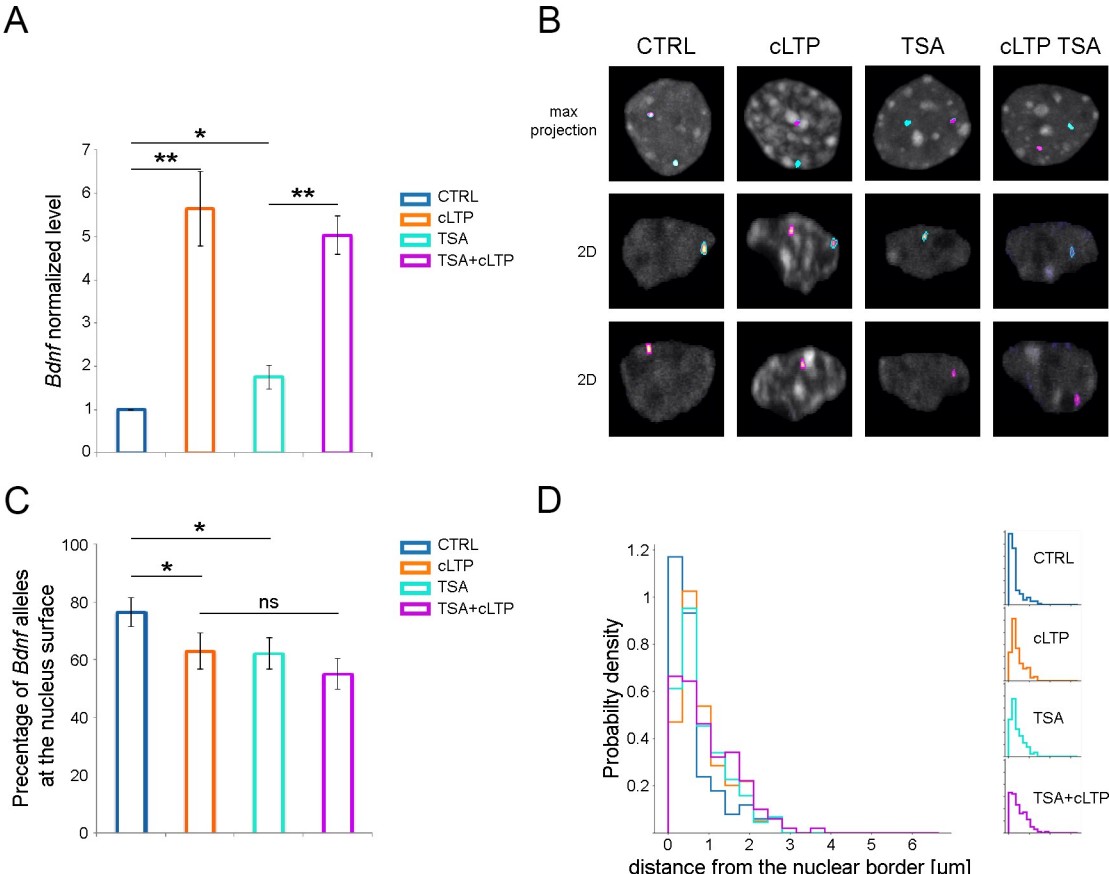

**Fig 4. The inhibition of histone deacetylases induces *Bdnf* transcription and repositioning independently from the neuronal stimulation.** (A) The graph shows the expression of *Bdnf* normalized to control in the hippocampal neurons incubated for 2 hours with DMSO vehicle (CTRL, blue bar) or picrotoxin, forskolin, and rolipram (cLTP, orange bar), incubated for 12 hours with TSA and 2 hours with DMSO (TSA, cyan bar) or picrotoxin, forskolin, and rolipram (TSA+cLTP, magenta bar). Kruskal-Wallis group comparison p<0.01, and the One-Sample T-test or Welch Two Sample t-test for pairwise comparison: * p<0.05, **p<0.01, *** p<0.001; error bars indicate standard error of the mean for 5 independent experiments (B) Representative pictures of the nuclei of hippocampal neurons treated as described above. Hoechst 3342 staining for chromatin is shown in greyscale and segmentation of FISH signals for the *Bdnf* gene are shown in magenta and cyan. (C) Percentages of nuclei with the minimum distance between the respective alleles and nucleus surface < 350 nm are shown (Chi-square test, all groups, p<0.01; Fisher's exact tests * p<0.05, *** p<0.001, error bars indicate standard deviation of the binomial distribution). (D) Quantitative analysis of the intracellular positions of *Bdnf* alleles in the nuclei of hippocampal neurons treated and color-coded as in A. The minimal distance between the respective alleles and nucleus surface is presented in the normalized histogram.

had no effect on *Bdnf* transcription (Fig 5A) and induced *Bdnf* allele repositioning only upon neuronal stimulation (Figs 5B–5D and S4). These results show again the causal relation between transcriptional activation of *Bdnf* and *Bdnf* allele's repositioning towards the nucleus center.

## Discussion

The three-dimensional organization of chromatin in the cell nucleus exhibits a higher-order regulation of gene expression [50–52]. The role of the nuclear lamina as a compartment regulating transcriptional activity is already quite well explored [53–55]. However, the knowledge of particular mechanisms, which are responsible for driving detachment of the genes from the nuclear lamina, is still vague [56].

To study the relationship between *Bdnf* repositioning, and morphological and clinical epileptic traits, we used the kainate model of epilepsy. This particular animal model was chosen

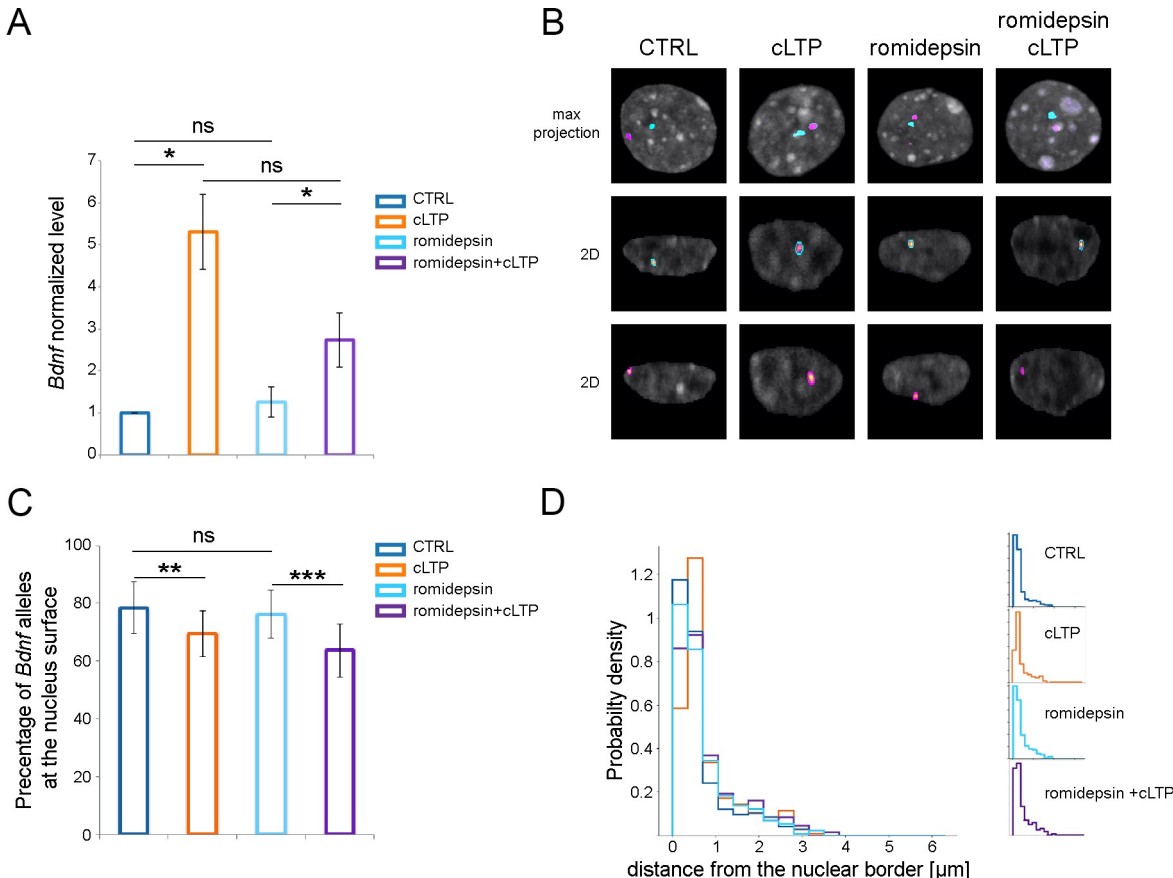

**Fig 5. The inhibition of HDAC1/2 is not sufficient to induce *Bdnf* transcription and repositioning.** (A) The graph shows the expression of *Bdnf* relative to *Gapdh* (normalized to control *Bdnf* level) in the hippocampal neurons incubated for 2 hours with DMSO vehicle (CTRL, blue bar) or picrotoxin, forskolin, and rolipram (cLTP, orange bar), incubated for 2 hours with 250 nM romidepsin and 2 hours with DMSO (romidepsin, light blue bar) or picrotoxin, forskolin, and rolipram (romidepsin+cLTP, violet bar). Kruskal-Wallis test for group comparison: p<0.01, One-Sample t-test or Welch two sample t-test for pairwise comparison: * p<0.05, *** p<0.001; error bars indicate standard error of the mean for 3 independent experiments (B) Representative picture of the nuclei of hippocampal neurons treated as described above. Hoechst 3342 staining for chromatin is shown in greyscale and segmentation of FISH signals for the *Bdnf* gene are shown in magenta and cyan. (C) Percentages of nuclei with the minimum distance between the respective alleles and nucleus surface < 350 nm are shown (Chi-square test, all groups, p<0.0001; Fisher's exact tests: ** p<0.01, *** p<0.001, error bars indicate standard deviation of the binomial distribution). (D) Quantitative analysis of the intracellular positions of *Bdnf* alleles in the nuclei of hippocampal neurons treated and color-coded as in A. The minimal distance between the respective alleles and nucleus surface is presented in the normalized histogram.

because it resembles morphological traits of TLE such as mossy fiber sprouting in the dentate gyrus and neurodegeneration in CA1 and 3 regions of the hippocampus [57,58]. Further, to study mechanisms of *Bdnf* allele repositioning, we used *in vitro* model of neuronal stimulation. This model was chosen to avoid problems with the distribution of the Actinomycin D [59] through the blood-brain barrier and to decrease the number of sacrificed animals. Transcriptomic studies by Dabrowski and collaborators showed that the expression pattern in excited neurons *in vivo* is sustained in *in vitro* model [60]. Moreover, Szepesi and collaborators (2013) [43] showed that the induction of excitation by rolipram, forskolin, and picrotoxin induced a formation of dendritic spine protrusions, which resembled *in vivo* formation of new synapses.

The nuclear lamina is known as a transcriptionally repressive nuclear compartment [61]. Williams et al. (2007) [62] and Peric-Hupces et al. (2010) [63] showed a rearrangement of interactions between chromatin and the nuclear lamina, and an association of such phenomenon

with transcription during differentiation of embryonic stem cells into neurons. However, the number of reports investigating a relationship between the rearrangement of chromatin architecture and transcription in fully differentiated neurons is very limited. In the presented study, using *in vitro* model, we confirmed our previous finding [34], where we showed that after neuronal excitation *Bdnf* alleles had repositioned toward the nuclear center. However, in comparison to the aforementioned paper, in the *in vitro* model, the percentage of alleles localized at the nuclear periphery was higher in the control cells, and the difference to stimulated neurons was smaller than in *in vivo* studies. Such discrepancy might be a result of a more uniform environment in cell culture, compared to *in vivo* situations, where spontaneous neuronal activity is present. Additionally, the kainate treatment, used in the *in vivo* studies, induces much stronger neuronal excitation than chemically induced LTP applied *in vitro*. In the presented study we showed that inhibition of transcription prevents *Bdnf* repositioning in neuronal nuclei after stimulation. It suggests that depolarization of the neuronal membrane is not sufficient for the detachment of *Bdnf* alleles, but the transcriptional activity itself is necessary for full repositioning. Our finding is consistent with the expertise presented by Crepaldi and colleagues (2013), who showed that in cultured cortical neurons depolarization induced by KCl stimulation evoked repositioning of activity-induced genes, including *Bdnf*, into transcription factories [33]. Moreover, this study showed that repositioning is controlled by transcription factor TFIIIC. Our current finding complements the previous results [34], where we showed the association of *Bdnf* repositioning and changes in transcription level during epileptogenesis. Both of those studies support the hypothesis that *Bdnf* allele repositioning acts as a kind of molecular memory of the cell to prepare neurons for future activation. The finding is also in agreement with the report by Ito and colleagues (2014) showing, that loss of three-dimensional architecture in neuronal nuclei leads to impaired transcription of several genes [64].

Surely, the presented mechanism is not the only one involved in the *Bdnf* repositioning. Presumably, several pathways must be orchestrated to activate *Bdnf* transcription and detachment of gene from the nuclear lamina, as well as its reposition toward the nuclear center. One possible mechanism may involve cohesins, which are well-known genome organizers [65] (and references therein). In Cornelia de Lange Syndrome, which belongs to cohesinopathies and is associated with epileptic seizures [66,67] acetylation of cohesins is impaired due to mutation in the gene encoding histone deacetylase HDAC8 [68]. Our results obtained with the use of inhibitors of histone deacetylases speak for this scenario. It has been shown that HDAC inhibitors induce hyperacetylation of more than 1700 proteins, which included chromatin remodelers, transcription factors, and protein kinases [69]. The trichostatin A- less specific but potent HDAC inhibitor, induced *Bdnf* transcription and repositioning even in the absence of neuronal activation. On the contrary, the more selective romidepsin, which inhibits mainly HDAC1 and 2, had no impact on both processes in silent neurons and upon stimulation. The cohesin-dependent mechanism may be at least in part responsible for *Bdnf* transcriptional activation, detachment, and repositioning, as CTCF, which acts in concert with cohesins [70] was shown to regulate the transcription of *Bdnf* [71].

Also, transcription factors such as Serum Response Factor (SRF) seem to be good candidates responsible for *Bdnf* repositioning. It is known that SRF can regulate *Bdnf* transcription [72] and its deletion leads to increased epileptogenesis and differential expression of more than 370 activity-induced genes including reduced transcription of *Bdnf* [73]. The disturbed balance of expression of inhibitory/excitatory regulators may contribute to the increased epileptogenesis, which was manifested by more frequent and acute spontaneous seizures in KO animals.

Furthermore, it was shown, that histone acetylation at the promoter region is necessary for activation of gene expression [74]. Histone deacetylases reverse this process and are taking part in creating a repressive zone by the nuclear lamina [75]. In a presented study we showed,

that inhibition of histone deacetylases with paninhibitor Trichostatnin A leads to *Bdnf* transcription and repositioning. Similar activation of *Bdnf* gene transcription upon TSA treatment was shown for the Hek293 cell line [76]. Also, HDACs' involvement in *Bdnf* expression is well established [28,48,77–80]. However, the exact mechanisms of HDACs' participation in the higher-order mechanisms of transcriptional regulation of BDNF are unknown. One possible mechanism might involve Methyl-CpG-Binding protein 2 (MeCP2). Activation of neurons leads to the decreased CpG methylation of the *Bdnf* promoter region and dissociation of the MeCP2-mSin3A-HDAC1 silencing complex [25]. Besides, MeCP2, which expression is reduced upon TSA treatment [81], binds to the inner nuclear membrane lamin B receptor [82]. Therefore the association of transcriptionally inactive *Bdnf* to the nuclear lamina and its detachment upon TSA treatment may be dependent on MeCP2 protein. Mutations of the MeCP2 encoding gene are responsible for most cases of Rett Syndrome, a neurodevelopmental disorder, in which patients develop seizures [83].

Additionally, the process of *Bdnf* repositioning might involve actin-based molecular motors, since studies by Serebryannyy et al. (2016) [84] reported that actin regulates the function of HDAC1 and HDAC2 and also can be involved in gene expression by association with RNA polymerase II.

Finally, in the presented study we attempted to investigate the involvement of *Bdnf* repositioning in the pathogenesis of TLE in rats. We observed that the percentage of nuclei with the *Bdnf* allele at the nuclear periphery is negatively correlated with the intensity of mossy fiber sprouting. This observation is consistent with an idea that BDNF takes part in sprouting events in epilepsy [17,18]. In addition the intensity of sprouting correlated positively with the increased level of aggression, which is one of the behavioral traits typical for TLE [85]. The Actinomycin D, which has been already used in treatments of several types of cancer [86–88], has blocked *Bdnf* repositioning. However, the potential use of Actinomycin D for the treatment of neurological disorders including epilepsy would be challenging due to difficulties of its distribution through the blood-brain barrier [59]. Taking into account all the results together, the presented study supports the idea of Simonato [89], that *Bdnf* can be a very good target for novel anti-epileptic therapies.

Conclusively, the presented results are consistent with the current trend in research on the pathogenesis of epilepsy and show that neuronal cell nuclei are interesting targets to search for mechanisms of epileptogenesis.

## Supporting information

**S1 Fig. The representative images of mossy fiber sprouting and *Bdnf* alleles positions within neuronal nuclei in the TLE model.** Mossy fiber sprouting was verified by immunofluorescent staining for synaptoporin (left panels, depicted in green) in the molecular layer of DG region of the hippocampus. Representative pictures from animals at 4 weeks after administration of kainate are shown. The right panels show images of nuclei of granular neurons acquired from the same animals. Hoechst 3342 staining for chromatin is shown in a grey scale and segmentation of FISH signals for *Bdnf* gene are shown in magenta and cyan.
(TIF)

**S2 Fig. The causal relationship between *Bdnf* transcriptional activity and *Bdnf's* allele repositioning.** The representative pictures of the nuclei of hippocampal neurons incubated for 2 hours with DMSO vehicle (CTRL) or picrotoxin, forskolin, and rolipram (cLTP), incubated for 2 hours with Actinomycin D and 2 hours with DMSO (ActD) or picrotoxin, forskolin, and rolipram (ActD+cLTP). Hoechst 3342 staining for chromatin is shown in a grey scale and

segmentation of FISH signals for *Bdnf* gene are shown in magenta and cyan.
(TIF)

**S3 Fig. The inhibition of histone deacetylases induces *Bdnf* repositioning independently from the neuronal stimulation.** The representative pictures of the nuclei of hippocampal neurons incubated for 2 hours with DMSO vehicle (CTRL) or picrotoxin, forskolin, and rolipram (cLTP), incubated for 12 hours with Trichostatin A and 2 hours with DMSO (TSA) or picrotoxin, forskolin, and rolipram (cLTP TSA). Hoechst 3342 staining for chromatin is shown in a grey scale and segmentation of FISH signals for *Bdnf* gene are shown in magenta and cyan.
(TIF)

**S4 Fig. The inhibition of HDAC1/2 is not sufficient to induce *Bdnf* repositioning.** The representative pictures of the nuclei of hippocampal neurons incubated for 2 hours with DMSO vehicle (CTRL) or picrotoxin, forskolin, and rolipram (cLTP), incubated for 2 hours with romidepsin and 2 hours with DMSO (romidepsin) or picrotoxin, forskolin, and rolipram (cLTP romidepsin). Hoechst 3342 staining for chromatin is shown in a grey scale and segmentation of FISH signals for *Bdnf* gene are shown in magenta and cyan.
(TIF)

**S1 File. Clinical correlates and statistical analysis for Figs 1–5.**
(XLSX)

## Author Contributions

**Conceptualization:** Agnieszka Walczak, Grzegorz Marek Wilczynski, Joanna Dzwonek, Adriana Magalska.

**Data curation:** Grzegorz Marek Wilczynski, Adriana Magalska.

**Formal analysis:** Anna Skupien-Jaroszek, Agnieszka Walczak, Iwona Czaban, Katarzyna Karolina Pels, Błażej Ruszczycki, Joanna Dzwonek, Adriana Magalska.

**Funding acquisition:** Grzegorz Marek Wilczynski, Joanna Dzwonek, Adriana Magalska.

**Investigation:** Anna Skupien-Jaroszek, Agnieszka Walczak, Iwona Czaban, Andrzej Antoni Szczepankiewicz, Joanna Dzwonek.

**Methodology:** Anna Skupien-Jaroszek, Katarzyna Karolina Pels, Katarzyna Krawczyk.

**Project administration:** Grzegorz Marek Wilczynski.

**Supervision:** Grzegorz Marek Wilczynski, Joanna Dzwonek, Adriana Magalska.

**Validation:** Iwona Czaban, Adriana Magalska.

**Visualization:** Anna Skupien-Jaroszek, Agnieszka Walczak, Adriana Magalska.

**Writing – original draft:** Agnieszka Walczak, Adriana Magalska.

**Writing – review & editing:** Grzegorz Marek Wilczynski, Joanna Dzwonek, Adriana Magalska.

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
