## [Decision Letter · Decision Letter 0]

5 Nov 2020

PONE-D-20-26926

An important role of the interplay between Bdnf transcription and histone acetylation in epileptogenesis

PLOS ONE

Dear Dr. Magalska,

Thank you for submitting your manuscript to PLOS ONE. After careful consideration, we feel that it has merit but does not fully meet PLOS ONE’s publication criteria as it currently stands. Therefore, we invite you to submit a revised version of the manuscript that addresses the points raised during the review process.

The reviewers have raised several important points, all of which should be addressed in your revision. I agree with the reviewers that the most critical issues include: 

1. providing adequate images to support all the conclusions

2. using more specific HDAC inhibition strategies than TSA

3. carefully revising/editing the  text to make sure it accurately describes the data/conclusions

5. providing sufficient methodological details on quantitations and statistics. 

We look forward to receiving your revised manuscript.

Kind regards,

Michal Hetman

Academic Editor

PLOS ONE

Journal Requirements:

Reviewers' comments:

Reviewer's Responses to Questions

**Comments to the Author**

1. Is the manuscript technically sound, and do the data support the conclusions?

Reviewer #1: Partly

Reviewer #2: Partly

2. Has the statistical analysis been performed appropriately and rigorously? 

Reviewer #1: Yes

Reviewer #2: I Don't Know

3. Have the authors made all data underlying the findings in their manuscript fully available?

Reviewer #1: Yes

Reviewer #2: Yes

4. Is the manuscript presented in an intelligible fashion and written in standard English?

Reviewer #1: Yes

Reviewer #2: Yes

5. Review Comments to the Author

Reviewer #1: Magalska et al. aim to uncover if repositioning of Bdnf alleles from the nuclear periphery to the nuclear center after chemically stimulation in rat neurons is a consequence of or cause of transcriptional activation of the Bdnf gene. Two methods of stimulation were used: an in vivo approach where kainate-induced seizures were used as a validated method to induce neuronal Bdnf transcription and an in vitro method of cLTP in primary rat neurons which has also been shown to induce Bdnf transcription. Overall, the evidence presented is insufficient and sometimes disagrees with the authors’ conclusions. The authors are clearly capable of producing the necessary data and figures based upon their previous publication in this area of study however for this manuscript, evidence is lacking. Stronger evidence is needed that may include additional imaging in primary neurons and a more specific HDAC inhibitor coupled with isoform-specific genetic silencing to demonstrate a role for HDACs. In Figure 2, it is unclear why the authors did not provide the microscopy FISH images from which the conclusions in Figure 2 were drawn in the main text. Lastly, in Figure 3, the images used contradict the authors’ conclusion (see below). This is an interesting and potentially fruitful area of research and the authors may therefore be given the opportunity to extensively re-work this study.

Major Points:

1. Please provide additional images that were used for quantification in supplementary data for figures 2, 3, and 4.

2. No data is shown for the quantification Bdnf allele repositioning in response to kainate-induced epileptogenesis, only a final plot of correlation of mossy fiber sprouting with the allele repositioning. Readers are unable to see how you obtained these results (Figure 2).

3. Organization of text needs to be edited. For example, under the subheading Figure 1, authors mention panels in Figure 2. Please edit the flow of the text to be more logical by moving all subheadings for Results to be sequential in the first part of the results and grouping all figure legends at the end with the corresponding figure.

4. The paragraph following “The causal relationship between Bdnf allele transcriptional activity and repositioning”: these images do not support the authors’ conclusion that, “Presented results show that transcriptional activity is a cause of Bdnf repositioning.” While actinomycin D inhibited transcription of Bdnf, repositioning of the Bdnf allele away from the periphery and to the center of the nucleus still occurs in its presence according to the representative images used. Please provide additional images used for quantification; the ones currently used are confusing and do not support the authors’ argument. The authors report “Inhibition of transcription completely blocked

5. Bdnf repositioning upon cLTP treatment (Fig 3B-D).” which I can’t see from these images as one allele is clearly located away from the periphery, identical to the cLTP alone treatment.

6. Please justify the use of Welch’s ANOVA for non homogenous variance by providing the statistics readouts from GraphPad Prism in supplemental data.

Minor Points:

1. Page 3: Revise “It is known that 3’ of the protein-coding exon is spliced to one of the eight of 5’ untranslated exons.” Doesn’t make sense, missing a word.

2. Figure 2B: 25.5% should be 25%

3. Figure 3D and 4D: please plot raw values with error bars. The way the data is currently presented does not account for sample size.

4. Page 11: Revise “The role of the nuclear lamina as a transcriptional activity regulating the compartment is already quite well explored (52-54).”

5. Please continue to revise overall grammar and sentence structure, especially in the discussion.

Reviewer #2: In this manuscript, the authors focused on their previous findings regarding the intranuclear repositioning of BDNF allele induced by kainate-induced neuronal activation. The authors investigated the correlation between the intranuclear repositioning of BDNF allele and clinical/morphological traits of kainate-induced epilepsy in particular. Furthermore, the authors used in vitro hippocampal cultures to investigate mechanisms of the repositioning of BDNF allele. Although this is an interesting work showing the correlation between higher-order epigenetic regulation of BDNF gene and epileptogenesis, several concerns lessen my enthusiasm for this paper. I listed my major points below.

1. The title of this manuscript is unlikely to represent this work appropriately. I think that one of the important points in this manuscript would be the interplay between the repositioning of BDNF allele and the seizure status. The title should be revised in order to represent this work, if possible.

2. In Figure 1A, the intensity of seizures was assessed by modified 6-grade Racine’s scale (from 0 to 5-grade). However, only 2 grades (white (5-grade?) and gray (2-grade?)) were shown in a pie chart (Fig. 1A). Is it correct? This pie chart seems not to reflect the data shown in Table 1, and therefore, the chart should be explained carefully.

3. The authors estimated the extend of mossy fiber sprouting that was based on the intensity of immunostaining with an anti-synaptoporin antibody. However, I could not understand how the extend of mossy fiber sprouting was classified into 5-grade scale (from level 0 to 4). The quantification method should be described.

4. I think that the result regarding the negative correlation between BDNF allele at the nuclear periphery and level of mossy fiber sprouting (Fig. 2C) is quite interesting. Is a similar correlation between BDNF allele at the nuclear periphery and intensity of seizures (assessed by modified 6-grade Racine’s scale) observed? And also, is a similar correlation between level of mossy fiber sprouting and intensity of seizures observed?

5. I think that the representative images shown in Figure 3B do not reflect the data shown in Figure 3C and D. Namely, the images seem to indicate that actinomycin D reduced the percentage of BDNF allele at the nuclear periphery in both control and cLTP-induced neurons. Furthermore, the images shown in Figure 3B closely resemble the images shown in Figure 4B (almost the same). The authors should revise the manuscript very carefully.

6. The authors show that HDAC inhibition was sufficient for the repositioning of BDNF allele toward the nuclear center, and they described the possibility of the involvement of HDAC1, 2, and 8, in the intranuclear repositioning of BDNF allele. In this work, the authors used TSA to inhibit HDACs. However, TSA is a pan-HDAC inhibitor. The authors’ proposal would be strengthened if they use class-specific HDAC inhibitors.

(Minor points)

1. How long were the cells treated with TSA? (In Method section, the cells were treated with TSA for 12 h. However, in Results section, the cells were treated with TSA for 2 h. Which is correct time?)

2. In Results section (page 8), “Here we observed, that the level of sprouting correlated...” should be “...the level of sprouting negatively correlated...”.

3. In Discussion section, the authors mentioned that the percentage of BDNF allele localized at the nuclear periphery in control hippocampal cells was higher than that in vivo model. They discuss that “Such discrepancy might be a result of a more uniform environment in cell culture compared to in vivo situation”. Spontaneous neuronal activity in vivo situation would be stronger than that in vitro cultured neurons, and this may be another possible reason.

4. In Discussion section, the authors mentioned that SRF may be a candidate factor responsible for the repositioning of BDNF allele. However, SRF deletion in mice leads to increased epileptogenesis, despite decreased expression of BDNF gene is observed in the SRF-deleted mice brain. The results obtained using SRF-deleted mice seem to contradict a series of previous reports showing that BDNF is involved in seizures.

5. In the end of Discussion section, the authors described that “HDAC activity might be a good target for the treatment of TLE”. Does it mean that HDAC activation might be beneficial for the treatment of TLE? I think that this should be clarified more.

6. In Figures 3D and 4D, error bars should be added.

6. PLOS authors have the option to publish the peer review history of their article (what does this mean?). If published, this will include your full peer review and any attached files.

Reviewer #1: **Yes: **Claes Wahlestedt

Reviewer #2: No

---

## [Author Response · Author response to Decision Letter 0]

5 May 2021

We are very grateful for all the comments and suggestions, which helped us improve our paper. We have changed the main text of manuscript accordingly.

Our detailed response to the editor's and reviewers’ comments is as follows:

Editor:

1. providing adequate images to support all the conclusions 

The mistaken images were corrected and additional figures are presented in the supplementary materials. 

2. using more specific HDAC inhibition strategies than TSA 

Following the Editor’s and Reviewer’s comments, we have performed additional in vitro experiments with the use of romidepsin, a selective inhibitor of HDAC1 and 2. The results are presented in Fig.5 and Supplementary Fig.S5. For more details please see the response to the Rev.1 general comment and Rev.2 comment 6.

3. carefully revising/editing the text to make sure it accurately describes the data/conclusions 

The text has been corrected accordingly.

5. providing sufficient methodological details on quantitations and statistics.

The detailed description of quantification and statistical analysis was included in the supplementary S1 file and in the Methods section.

Reviewer 1:

Magalska et al. aim to uncover if repositioning of Bdnf alleles from the nuclear periphery to the nuclear center after chemically stimulation in rat neurons is a consequence of or cause of transcriptional activation of the Bdnf gene. Two methods of stimulation were used: an in vivo approach where kainate-induced seizures were used as a validated method to induce neuronal Bdnf transcription and an in vitro method of cLTP in primary rat neurons which has also been shown to induce Bdnf transcription. Overall, the evidence presented is insufficient and sometimes disagrees with the authors’ conclusions. The authors are clearly capable of producing the necessary data and figures based upon their previous publication in this area of study however for this manuscript, evidence is lacking. 

Stronger evidence is needed that may include additional imaging in primary neurons and a more specific HDAC inhibitor coupled with isoform-specific genetic silencing to demonstrate a role for HDACs. In Figure 2, it is unclear why the authors did not provide the microscopy FISH images from which the conclusions in Figure 2 were drawn in the main text. Lastly, in Figure 3, the images used contradict the authors’ conclusion (see below). This is an interesting and potentially fruitful area of research and the authors may therefore be given the opportunity to extensively re-work this study.

In agreement with the Reviewer’s comment, we performed additional in vitro experiments with the use of romidepsin which is an inhibitor of HDAC 1 and 2. The results are presented in Fig.5 and S5. Appropriate fragments are inserted into the Methods (line 130), the Results (lines 290-293), the Figures’ Legends (lines 314-329 and 682-687) and the Discussion (line 386-387) sections. Please find response to all other concerns below.

Major Concerns

1. Please provide additional images that were used for quantification in supplementary data for figures 2, 3, and 4.

Addressing the reviewer’s comment, we have provided additional images in Supplementary figures S2, S3, S4 and S5, describing experimental results shown on Figures 2, 3, 4, and a new Fig. 5 respectively. 

2. No data is shown for the quantification Bdnf allele repositioning in response to kainate-induced epileptogenesis, only a final plot of correlation of mossy fiber sprouting with the allele repositioning. Readers are unable to see how you obtained these results (Figure 2).

Following the Reviewer’s remark, we have included the data used for the quantification of correlation in the Supplementary S1 file. The distance of BDNF alleles from the nuclear border was measured on 3D confocal images with the use of Segmentation Magic software, developed in our Laboratory (for further details please read: Ruszczycki, B.; Pels, K.K.; Walczak, A.; Zamłyńska, K.; Such, M.; Szczepankiewicz, A.A.; Hall, M.H.; Magalska, A.; Magnowska, M.; Wolny, A.; et al. Three-Dimensional Segmentation and Reconstruction of Neuronal Nuclei in Confocal Microscopic Images. Front. Neuroanat. 2019, 13, 81, doi:10.3389/fnana.2019.00081.).

The level of sprouting was measured using Image J, defined as a mean fluorescence intensity of synaptoporin immunostaining of the same region of interest in the molecular layer of dentate gyrus. Then, the intensity was ranked from 0 (minimal staining comparable with the untreated animals) to 4- the highest intensity in KA treated animals. The results are presented in the Table 1 (shown also below) and in the supplementary S1 file. 

Animal ID selected area in pixels the mean synaptoporin intensity the sprouting level

1B 10207,201 10,127 2

2B 10009,387 11,571 2

1G 9940,673 30,554 4

2G 10325,890 16,044 3

3G 10197,368 4,345 0

4G 10523,704 25,694 4

1C 10276,725 4,198 0

2C 10506,294 5,151 1

3R 9447,584 5,674 1

4R 10484,141 5,444 1

And intensity scores:

The mean synaptoporin intensity scale (0-255 grey scale) The level of sprouting

0 to 4 0

5 to 9 1

10 to 15 2

16 to 20 3

above 21 4

Additionally two independent observers estimated sprouting visually in the same brain area directly using a microscope. The sprouting scores and correlation scores are included in supplementary S1 file.

In the Supplementary Figure S2, we have included the representative images for each analyzed animal of the synaptoporin immunostaining and BDNF alleles positioning within the cell nuclei.

3. Organization of text needs to be edited. For example, under the subheading Figure 1, authors mention panels in Figure 2. Please edit the flow of the text to be more logical by moving all subheadings for Results to be sequential in the first part of the results and grouping all figure legends at the end with the corresponding figure.

The text was corrected following the Reviewer’s comment.

4. The paragraph following “The causal relationship between Bdnf allele transcriptional activity and repositioning”: these images do not support the authors’ conclusion that, “Presented results show that transcriptional activity is a cause of Bdnf repositioning.” While actinomycin D inhibited transcription of Bdnf, repositioning of the Bdnf allele away from the periphery and to the center of the nucleus still occurs in its presence according to the representative images used. Please provide additional images used for quantification; the ones currently used are confusing and do not support the authors’ argument. The authors report “Inhibition of transcription completely blocked

And 

5. Bdnf repositioning upon cLTP treatment (Fig 3B-D).” which I can’t see from these images as one allele is clearly located away from the periphery, identical to the cLTP alone treatment.

The images representing localization of BDNF alleles in cell nuclei were swapped by mistake. Responding to the Reviewer’s comment, we have corrected the figures by introducing proper images. Additionally, in the Supplementary Figures S3-S5, we have included more representative images of nuclei used for the calculations, for all the treatments.

6. Please justify the use of Welch’s ANOVA for non homogenous variance by providing the statistics readouts from GraphPad Prism in supplemental data.

In the previous version of the manuscript, we used Welch’s ANOVA statistical analysis, since the variances calculated with Bartlet test were not homogenous. In the present version of the manuscript improved, after addressing Reviewer’s remarks, we have included new data for more specific histone deacetylases inhibitor, romidepsin. We decided to normalize data to the control (DMSO treatment) to compare the obtained data from the all in vitro experiments. Therefore, we used Kruskal-Wallis test for group comparisons and One Sample T-test for comparison whether the normalized value differs from the unity, or Welch Two sample T-test for pairwise comparisons between treatment groups. All raw data and results of statistical analysis are shown in supplementary S1 file.

Minor Points:

7. Page 3: Revise “It is known that 3’ of the protein-coding exon is spliced to one of the eight of 5’ untranslated exons.” Doesn’t make sense, missing a word.

The text was corrected to: It is known that 3’ end of the protein-coding exon is spliced to one of the eight of 5’ exons, which are controlled by their unique promoters (20-22).

8. Figure 2B: 25.5% should be 25%

To fulfill the Reviewer’s comments, we have reanalyzed the data obtained from the KA treated animals and increased the number of animals to 10. Therefore all the percentages had changed. 

9. Figure 3D and 4D: please plot raw values with error bars. The way the data is currently presented does not account for sample size.

Addressing the Reviewer’s remark, we introduced the error bars in the Figures 2,3-4 and in the new Fig.5. All raw data used for statistical analysis are shown in supplementary S1 file. 

10. Page 11: Revise “The role of the nuclear lamina as a transcriptional activity regulating the compartment is already quite well explored (52-54).”

We corrected the sentence to: The role of the nuclear lamina as a compartment regulating the transcriptional activity is already quite well explored (52-54).

11. Please continue to revise overall grammar and sentence structure, especially in the discussion.

Following the Reviewer’s suggestion, the text was subjected to extensive language corrections.

Reviewer #2:

In this manuscript, the authors focused on their previous findings regarding the intranuclear repositioning of BDNF allele induced by kainate-induced neuronal activation. The authors investigated the correlation between the intranuclear repositioning of BDNF allele and clinical/morphological traits of kainate-induced epilepsy in particular. Furthermore, the authors used in vitro hippocampal cultures to investigate mechanisms of the repositioning of BDNF allele. Although this is an interesting work showing the correlation between higher-order epigenetic regulation of BDNF gene and epileptogenesis, several concerns lessen my enthusiasm for this paper. I listed my major points below.

Major Points

1. The title of this manuscript is unlikely to represent this work appropriately. I think that one of the important points in this manuscript would be the interplay between the repositioning of BDNF allele and the seizure status. The title should be revised in order to represent this work, if possible.

In agreement with the Reviewer’s comment, we modified the title to: “The interplay of seizures-induced axonal sprouting and transcription-dependent Bdnf repositioning in the model of temporal lobe epilepsy”

2. In Figure 1A, the intensity of seizures was assessed by modified 6-grade Racine’s scale (from 0 to 5-grade). However, only 2 grades (white (5-grade?) and gray (2-grade?)) were shown in a pie chart (Fig. 1A). Is it correct? This pie chart seems not to reflect the data shown in Table 1, and therefore, the chart should be explained carefully.

Addressing the Reviewer’s comments, we have revised the results carefully. We have repeated the synaptoporin immunostaining and included animals 2C and 4G, previously excluded from the analysis because of technical problems with synaptoporin staining. The animals observed in our kainate experiments developed seizures of grade 2, 4 and 5, in the agreement with the data presented below, Table 1 and supplementary S1 file.

 Animal ID seizures level

1B I blue 2

1C I black 5

1G I green 5

2B II blue 5

2C II black 4

2G II green 5

3G III green 2

3R III red 5

4G IV green 5

4R IV red 2

3. The authors estimated the extend of mossy fiber sprouting that was based on the intensity of immunostaining with an anti-synaptoporin antibody. However, I could not understand how the extend of mossy fiber sprouting was classified into 5-grade scale (from level 0 to 4). The quantification method should be described.

The intensity of immunostaining was calculated with the use of ImageJ. Then, the mean fluorescence intensity was assessed using 5-grade scale, describing the level of sprouting according to the intervals presented in the following Tables, that were also included in the supplementary S1 file

The mean synaptoporin intensity scale (0-255 grey scale) The level of sprouting

0 to 4 0

5 to 9 1

10 to 15 2

16 to 20 3

above 21 4

Animal ID selected area in pixels the mean synaptoporin intensity the sprouting level

1B 10207,201 10,127 2

2B 10009,387 11,571 2

1G 9940,673 30,554 4

2G 10325,890 16,044 3

3G 10197,368 4,345 0

4G 10523,704 25,694 4

1C 10276,725 4,198 0

2C 10506,294 5,151 1

3R 9447,584 5,674 1

4R 10484,141 5,444 1

Additionally, two independent observers estimated sprouting visually in the same brain area directly using a microscope. The sprouting scores and correlation scores are included in supplementary S1 file.

4. I think that the result regarding the negative correlation between BDNF allele at the nuclear periphery and level of mossy fiber sprouting (Fig. 2C) is quite interesting. Is a similar correlation between BDNF allele at the nuclear periphery and intensity of seizures (assessed by modified 6-grade Racine’s scale) observed? And also, is a similar correlation between level of mossy fiber sprouting and intensity of seizures observed?

We have calculated correlations suggested by the Reviewer including additional animals and newly obtained data. The calculated Pearson’s correlations are enlisted below and included in supplementary S1 file. We have found one additional significant positive correlation between the level of sprouting and aggression. The increase in the aggressive behavior has frequently been reported in association with the development of temporal lobe epilepsy. However no significant correlation was found between the intensity of initial seizures and sprouting or seizure and the percentage of nuclei with the Bdnf alleles localized on the nuclear periphery. We decided to show correlations for sprouting level measured with Image J and estimated subjectively by two independent observers as obtained results were very similar. 

Correlation: The sprouting level Image J vs % of nuclei with Bdnf loci on the perifery

(Pearson R= -0.6747943402484425, p-value= 0.03230502682762817)

Correlation: The subjective Sprouting vs % of nuclei with Bdnf loci on the perifery

(Pearson R= -0.7086823744896321, p-value= 0.02177595135907097)

Correlation: The Sprouting level Image J vs agression

(Pearson R=0.7460317460317462, p-value= 0.01321972976110183

Correlation: % of nuclei with Bdnf loci on the perifery vs Seizures 

(Pearson R= -0.5272596866202862, p-value= 0.11731880152677869)

Correlation: The sprouting level vs Seizures

(Pearson R= 0.4259177099999599, p-value= 0.2197183529541219)

5. I think that the representative images shown in Figure 3B do not reflect the data shown in Figure 3C and D. Namely, the images seem to indicate that actinomycin D reduced the percentage of BDNF allele at the nuclear periphery in both control and cLTP-induced neurons. Furthermore, the images shown in Figure 3B closely resemble the images shown in Figure 4B (almost the same). The authors should revise the manuscript very carefully.

The images representing localization of BDNF alleles in cell nuclei were swapped by mistake. In agreement with the Reviewer’s comment, we corrected the figures by introducing proper images. Additionally, we have included more representative images of nuclei used for the calculations in supplementary figures S3 and S4.

6. The authors show that HDAC inhibition was sufficient for the repositioning of BDNF allele toward the nuclear center, and they described the possibility of the involvement of HDAC1, 2, and 8, in the intranuclear repositioning of BDNF allele. In this work, the authors used TSA to inhibit HDACs. However, TSA is a pan-HDAC inhibitor. The authors’ proposal would be strengthened if they use class-specific HDAC inhibitors.

In agreement with the Reviewer’s criticism, we performed additional in vitro experiments with the use of romidepsin which is an inhibitor of HDAC 1 and 2. The results are presented in Fig.5 and S5. Appropriate fragments are inserted into the Methods (line 130), the Results (lines 290-293), the Figures’ Legends (lines 314-329 and 682-687) and the Discussion (line 386-387) sections.

(Minor points)1. How long were the cells treated with TSA? (In Method section, the cells were treated with TSA for 12 h. However, in Results section, the cells were treated with TSA for 2 h. Which is correct time?)

Cells were incubated with TSA for 12 h. The mistake in the text was corrected.

1. In Results section (page 8), “Here we observed, that the level of sprouting correlated...” should be “...the level of sprouting negatively correlated...”.

The text was corrected to: “Here we observed, that the level of sprouting negatively correlated (R= -0.67, Pearson correlation) with the percentage of the nuclei with Bdnf allele localized at the nuclear periphery (Fig. 2 C).”

2. In Discussion section, the authors mentioned that the percentage of BDNF allele localized at the nuclear periphery in control hippocampal cells was higher than that in vivo model. They discuss that “Such discrepancy might be a result of a more uniform environment in cell culture compared to in vivo situation”. Spontaneous neuronal activity in vivo situation would be stronger than that in vitro cultured neurons, and this may be another possible reason.

Thank you for a great comment, We have included it in the text, which now sounds:

“Such discrepancy might be a result of more uniform environment in cell culture, compared to in vivo situation, where spontaneous neuronal activity is present.”

3. In Discussion section, the authors mentioned that SRF may be a candidate factor responsible for the repositioning of BDNF allele. However, SRF deletion in mice leads to increased epileptogenesis, despite decreased expression of BDNF gene is observed in the SRF-deleted mice brain. The results obtained using SRF-deleted mice seem to contradict a series of previous reports showing that BDNF is involved in seizures.

Thank you for this interesting comment. In the mentioned publication of Kuzniewska et al. SRF deletion induced differential expression of more than 370 activity-induced genes. These included regulators of both inhibitory and excitatory pathways. Therefore the increased epileptogenesis is rather a result of an imbalance between the inhibitory and excitatory pathways, than a reduced level of BDNF observed in SRF KO animals.

We have changed this part of the Discussion to explain better the results obtained by Kuzniewska et al.

5. In the end of Discussion section, the authors described that “HDAC activity might be a good target for the treatment of TLE”. Does it mean that HDAC activation might be beneficial for the treatment of TLE? I think that this should be clarified more.

We have removed this statement from the Discussion as our experiments with a selective inhibitor of HDAC1 and 2 showed no change in Bdnf transcription and positioning neither in silent nor activated neurons. The trichostatin A, which inhibits a broad spectrum of HDACs might induce Bdnf transcription and repositioning via chromatin silencer- MeCP2, which reduced levels are observed upon TSA treatment. We have included this possibility in the Discussion. 

6. In Figures 3D and 4D, error bars should be added.

Following the Reviewer’s remark, we introduced the error bars in the figures 3, 4 and in the new Fig.5. All data used for statistical analysis are shown in supplementary S1 file.

---

## [Decision Letter · Decision Letter 1]

18 May 2021

The interplay of seizures-induced axonal sprouting and transcription-dependent Bdnf repositioning in the model of temporal lobe epilepsy

PONE-D-20-26926R1

Dear Dr. Magalska,

We’re pleased to inform you that your manuscript has been judged scientifically suitable for publication and will be formally accepted for publication once it meets all outstanding technical requirements.

Kind regards,

Michal Hetman

Academic Editor

PLOS ONE

Additional Editor Comments (optional):

Reviewers' comments:

Reviewer's Responses to Questions

**Comments to the Author**

1. If the authors have adequately addressed your comments raised in a previous round of review and you feel that this manuscript is now acceptable for publication, you may indicate that here to bypass the “Comments to the Author” section, enter your conflict of interest statement in the “Confidential to Editor” section, and submit your "Accept" recommendation.

Reviewer #2: All comments have been addressed

2. Is the manuscript technically sound, and do the data support the conclusions?

Reviewer #2: Yes

3. Has the statistical analysis been performed appropriately and rigorously? 

Reviewer #2: Yes

4. Have the authors made all data underlying the findings in their manuscript fully available?

Reviewer #2: Yes

5. Is the manuscript presented in an intelligible fashion and written in standard English?

Reviewer #2: Yes

6. Review Comments to the Author

Reviewer #2: (No Response)

7. PLOS authors have the option to publish the peer review history of their article (what does this mean?). If published, this will include your full peer review and any attached files.

Reviewer #2: No

---

## [Editor Report · Acceptance letter]

24 May 2021

PONE-D-20-26926R1 

The interplay of seizures-induced axonal sprouting and transcription-dependent *Bdnf* repositioning in the model of temporal lobe epilepsy 

Dear Dr. Magalska:

I'm pleased to inform you that your manuscript has been deemed suitable for publication in PLOS ONE. Congratulations! Your manuscript is now with our production department. 

Kind regards, 

on behalf of

Dr. Michal Hetman 

Academic Editor

PLOS ONE